# Whole-genome sequencing of 128 camels across Asia reveals origin and migration of domestic Bactrian camels

Liang Ming et al.[#]

The domestic Bactrian camels were treated as one of the principal means of locomotion between the eastern and western cultures in history. However, whether they originated from East Asia or Central Asia remains elusive. To address this question, we perform whole-genome sequencing of 128 camels across Asia. The extant wild and domestic Bactrian camels show remarkable genetic divergence, as they were split from dromedaries. The wild Bactrian camels also contribute little to the ancestry of domestic ones, although they share close habitat in East Asia. Interestingly, among the domestic Bactrian camels, those from Iran exhibit the largest genetic distance and the earliest split from all others in the phylogeny, despite evident admixture between domestic Bactrian camels and dromedaries living in Central Asia. Taken together, our study support the Central Asian origin of domestic Bactrian camels, which were then immigrated eastward to Mongolia where native wild Bactrian camels inhabit.

[#]A full list of authors and their affiliations appears at the end of the paper.

Camels (*Camelus*, *Camelini*) contain two extant domestic species, the one-humped dromedary (*Camelus dromedarius*) and the two-humped Bactrian camel (*Camelus bactrianus*)[1,2]. Although the former herds are mainly feed in North Africa and West Asia, the latter herds live in the cold desert areas of Northeast and Central Asia. The wild Bactrian camel (*Camelus ferus*), the only representative of the wild tribe Camelini as a result of the extinction of the wild dromedary[3], is listed as critically endangered by the International Union for Conservation of Nature[4] with an estimation of a few hundreds to 2000 individuals[5,6]. Historically, the wild Bactrian camel was widely distributed throughout Asia, extending from the great bend of the Yellow River westward to central Kazakhstan (KAZA), but it can only be found in the Mongolian Gobi and the Chinese Taklimakan and Lop Noor deserts today[7]. Fossil and molecular evidence suggested that the ancestor of camels lived in North America and spread to Asia via the Bering land bridge around 11 or 16 million years ago[8,9]. Within the Camelini, the dromedaries and Bactrian camels were then split around 4 or 5 million years ago[9,10]. The domestication of camels, similar to many other innovations of domestic mammals such as horse-based transport[11], greatly promoted human mobility and represented a great leap forward for human civilization. For example, the Bactrian camels were rightfully considered as the principal means of locomotion across the bridge between the eastern and western cultures in the time of the Silk Road[12]. Today, they still serve as valuable sources of meat, milk, and wool to people's livelihoods in arid and semi-arid areas[1].

The origin of domestic dromedaries was recently revealed by world-wide sequencing of modern and ancient mitochondrial DNA (mtDNA), which suggested that they were at first domesticated in the southeast Arabian Peninsula[13]. However, the origin of domestic Bactrian camels remains controversial. One intuitive possibility was that the extant wild Bactrian camels were the progenitor of the domestic form, which were then dispersed from the Mongolian Plateau to the West gradually[7,12]. This hypothesis was supported by the presence of Camelid faunal remains at Neolithic sites near Mongolia (MG), although it was unclear these were the domestic as opposed to the wild ones[12]. Nevertheless, molecular studies based on mtDNAs[9,14] and Y chromosomes[15] discovered dramatic sequence variations between the wild and domestic Bactrian camels, suggesting that the extant wild Bactrian camel was a separate lineage[14]. Another possible place of origin was Iran (IRAN)[1], where early skeletal remains of domestic Bactrian camels (around 2500–3000 BC) were discovered[16]. Although prehistoric mtDNAs of Bactrian camels supported the idea that the domestication took place in Central Asia rather than in MG or East Asia[17], the incomplete archaeological findings and limited molecular markers provided little decisive information about the actual domestication history.

Whole-genome sequences contain much more molecular markers than mtDNAs, which were successfully used to portray the origin, migration, and admixture of humans[18–20] and domestic animals[21], such as dogs[22–25] and pigs[26–28]. The published genome assembly of Bactrian camels[10,29] provides a new opportunity to examine the evolutionary relationship between the extant wild and domestic Bactrian camels, and trace their origin. In this study, we perform whole-genome sequencing of 128 camels including both domestic and extant wild Bactrian camels from their typical habitats. We include dromedaries as well, because they are not only the outgroup of Bactrian camels in phylogeny but also have a long history of hybridization with Bactrian camels in breeding practice[2,30], especially in Central Asia. Our results support the Central Asian origin of domestic Bactrian camels and a roughly west-to-east route of migration back to the Mongolian Plateau.

## Results

### Sample collection and whole-genome sequencing.
A total of 105 domestic Bactrian camels across Asia, 19 wild Bactrian camels from Gobi-Altai region in MG, as well as 4 dromedaries from IRAN were gathered for this study (Supplementary Fig. 1 and Supplementary Table 1). The domestic Bactrian camels were chosen to cover as many major geographic regions as possible, including 55 from Inner MG (IMG), Xinjiang (XJ), and Qinghai of China, 28 from MG, 6 from KAZA, 10 from Russia (RUS) and 6 from IRAN. As a variety of local breeds were formed due to the wide utilization of domestic Bactrian camels in China and MG, eight different representative breeds were chosen from the regions. The other domestic Bactrian camels from Central Asia were living around the Caspian Sea.

After DNA extraction, individual genomes were sequenced to an average of 13× coverage (Supplementary Fig. 2 and Supplementary Table 2). The sequence reads were aligned to our previous genome assembly of the Bactrian camel[29] for variant calling. After stringent filtering (Supplementary Fig. 3), we totally identified 13.83 million single-nucleotide polymorphisms (SNPs) and 1.41 million small indels. Notably, the transition to transversion ratio of raw SNPs (2.29) was lower than that reported in dromedaries (2.31–2.34)[31], but it was increased to 2.44 after the filtering procedures, suggesting a quality improvement of identified SNPs. Functional annotation of the variants indicated that about 63.10% of them were intergenic, 33.62% were intronic, and 0.94% were exonic (Supplementary Table 3). There were 13.73 million, 6.39 million, and 10.55 million variants identified in the domestic Bactrian camels, wild Bactrian camels, and dromedaries, respectively. Although dromedaries were more divergent from both of the Bactrian camel species in phylogeny, the domestic Bactrian camels shared more variants with the dromedaries (66.73%) than with the wild Bactrian camel (39.31%) (Supplementary Fig. 4) due to the tremendous reduction of genetic variants observed in the extant wild Bactrian camel and to gene flow between dromedaries and domestic Bactrian camels. Among the domestic Bactrian camels, there were 12.68 million and 11.61 million variants identified in the East Asian and Central Asian populations, respectively (Supplementary Fig. 4). Although the domestic camels sampled from East Asia were more than those from Central Asia, the variant count private to each population showed no significant bias between the two areas (P-value = 0.77, two-tailed *t*-test; Supplementary Table 4).

### Genetic diversity and differentiation.
For a more detailed comparison of the genetic diversity among different populations, we first removed 14 individuals showing close genetic relationship with the remaining others (Supplementary Table 5). The pairwise nucleotide diversity $\pi$ (Fig. 1a) of dromedaries ($1.54 \times 10^{-3}$) was significantly higher than that of Bactrian camels from all geographic regions ($0.88 \times 10^{-3}$–$1.11 \times 10^{-3}$; Supplementary Table 6), which was in contrast to previous heterozygosity estimates based on individual genomes[10]. One important reason could be the hybridization practice between dromedaries and Bactrian camels in Central Asia[30]. Among the Bactrian camels, the wild population showed the lowest $\pi$ ($0.88 \times 10^{-3}$) compared with all of the domestic populations (Fig. 1a and Supplementary Table 6). Although this result violated many cases that wild animals usually have higher genetic diversity than their domestic counterparts such as dogs[24], pigs[27], and rabbits[32], it would happen in the case of endangered wild animals with an extremely small population size such as horses[33]. In addition, the domestic populations living in Central Asia generally showed a higher diversity ($1.03 \times 10^{-3}$–$1.11 \times 10^{-3}$) than those living in East Asia ($0.95 \times 10^{-3}$–$1.02 \times 10^{-3}$; Fig. 1a). The tendency was also

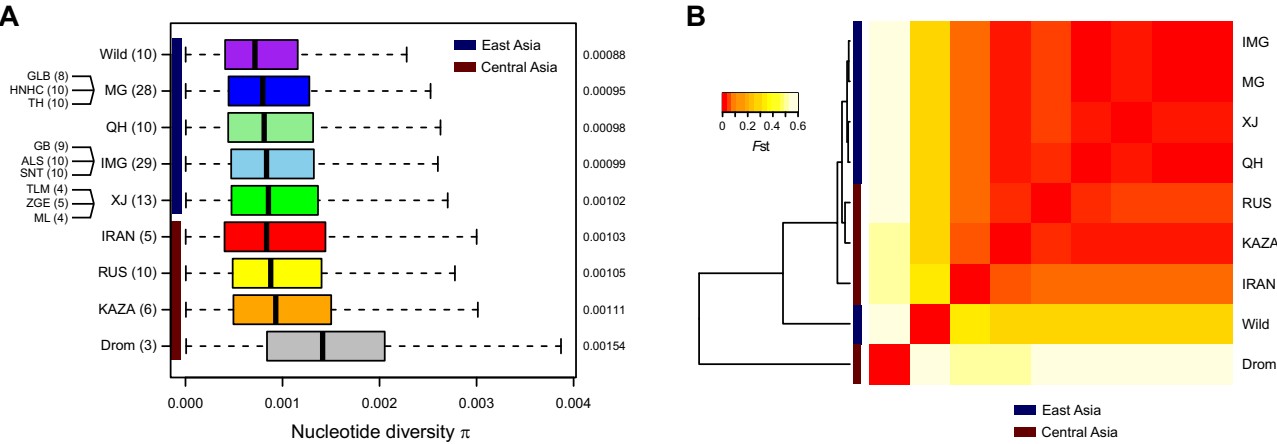

**Fig. 1 Genetic diversity and differentiation of the camel populations. a** Nucleotide diversity $\pi$. The boxplot shows $\pi$ for $2.0 \times 10^5$ 10 kb-sliding windows across the genome. The geographic origin and sample size of each population are shown on the left and the average of $\pi$ are shown on the right. Multiple local breeds were sampled for MG, IMG, and XJ. Individuals with close genetic relationships were removed. The boxplot elements are defined as follows: center line, median; box limits, the third and first quartiles; whiskers, 1.5 × interquartile range. **b** Pairwise population differentiation $F$st. The heatmap represents average $F$st for $2.0 \times 10^5$ 10 kb-sliding windows. The dendrogram represents hierarchical clustering of the populations based on $F$st.

confirmed by the Watterson's $\theta$ (Supplementary Fig. 5). Again, the hybridization with dromedaries in Central Asia could account for the higher diversity in the region.

We then measured the pairwise genetic distance between the camel populations by Weir's $F$st (Fig. 1b). The result was well in agreement with the known phylogeny, which indicated that the dromedaries had the highest $F$st with the Bactrian camels (0.54–0.64) and the wild Bactrian camels had the second highest $F$st with the domestic ones (0.27–0.31). The differentiation among the domestic Bactrian camels was much lower, in line with their recent single origin. Interestingly, among the domestic Bactrian camels, those from IRAN displayed the largest divergence with all others (0.05–0.06). To validate the population differentiation, we constructed a neighbor-joining (NJ) tree for all individuals based on their pairwise identity-by-state (IBS) matrix (Supplementary Fig. 6). The NJ tree also supported a monophyletic clade of all domestic Bactrian camels, within which IRAN formed the deepest branch.

A potential issue with the population genomic estimates was the reference bias, where using a single reference genome would lead to low efficiency in variant calling for individuals that highly differed from it[34]. To investigate the bias, we compared the missing count of variants among the three species, taking the sequencing depth as a covariate (Supplementary Table 7). The analysis of variance (ANOVA) showed that although the domestic Bactrian camels had no significant difference with the wild ones (P-value = 0.50), they indeed had lower missing count than dromedaries (P-value = $4.38 \times 10^{-3}$). To evaluate the impact of the bias on our estimates, we re-calculated the genetic diversity and $F$st with just synonymous SNPs (Supplementary Fig. 7), as coding sequences were more likely to be invariant across species. As a result, the estimates based on the synonymous SNPs were in good agreement with the whole genome for all species, suggesting that the reference bias had only minor effects on our population genomic estimates.

**Population structure with admixture**. To reveal the overall population structure with potential admixture, we pruned the SNPs by removing those with high linkage disequilibrium and potential functional effects. The multidimensional scaling (MDS) analysis based on the pruned subset reproduced the similar result as the full set (Fig. 2a and Supplementary Fig. 8). As expected, the

dromedaries and wild Bactrian camels could be separated by the first and second coordinates, respectively. When the third coordinate was incorporated in the MDS, IRAN was separated from all other domestic Bactrian camels (Fig. 2a).

To estimate different ancestral proportions, we performed population structure analysis with Admixture[35] by assuming $K$ ancestral populations (Fig. 2b). The cross-validation procedure supported that $K = 3$ was optimal (Supplementary Fig. 9), showing a clear division between the dromedaries, wild Bactrian camels, and domestic Bactrian camels. Evident introgression of domestic Bactrian camels into the Iranian dromedaries was observed, at least in one dromedary. Accordingly, the dromedary ancestry was prevalent in the Central Asian Bactrian camel populations including IRAN, KAZA, and RUS, with a proportion estimated as 1–10%. Moreover, we observed ancestry of domestic Bactrian camels in several wild individuals with a proportion of 7–15%. This could arise from ancestral polymorphisms, but it could also be caused by introgressive hybridization, which was observed with mtDNAs[36] and Y chromosomes[15], and was proposed to threaten the genetic distinctiveness of the wild species. Surprisingly, the wild camels contributed nearly nothing to the ancestry of domestic populations, even to the Mongolian populations, which share close habitats with the wild camels. Although most domestic Bactrian camels lacked differentiation when $K$ grew, IRAN was the first population to separate with a unique ancestry ($K = 5$; Fig. 2b).

As another method to examine the population structure with admixture, we inferred the population tree for the camels using TreeMix[37] (Fig. 2c). When no migration track was added, the tree topology again indicated that IRAN was the first population separated among all the domestic Bactrian camels (Supplementary Fig. 10). Increasing of migration tracks ($m = 1–3$) could greatly improve the fit of the model (Supplementary Fig. 11), which identified gene flows from dromedaries to domestic Bactrian camels in Central Asia, including KAZA, RUS, and IRAN with migration weights ranging from 4% to 9% (Supplementary Table 8). It was worth mentioning that although the migration track pointed at the end of the dromedary branch to IRAN, it pointed at the middle of the dromedary branch to KAZA and RUS (Fig. 2c). This could imply a ghost population related to the Iranian dromedary that contributed to the ancestry of KAZA and RUS. Additional migration track ($m = 4$) could

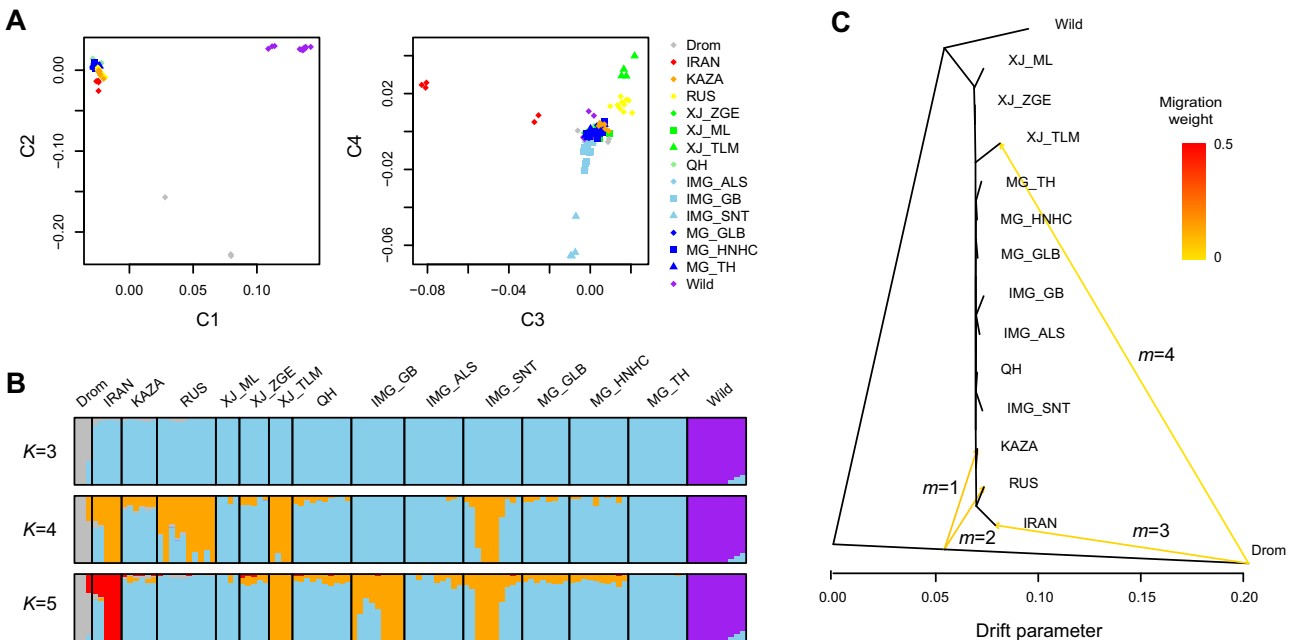

**Fig. 2 Population structure of the camels based on genome-wide SNPs. a** Multidimensional scaling (MDS) plot with coordinate 1–4 (C1–C4). **b** Admixture analysis assuming different number of ancestry K. The proportion of an individual's genome assigned to each ancestry is represented by different colors. **c** TreeMix analysis with different assumption of migration events m. The migration weight is the proportion of ancestry received from the donor population.

continue to improve the fit of the model, which indicated migration of the dromedary to a XJ breed (Fig. 2c). Although TreeMix detected no strong signal of migration between the wild and domestic Bactrian camels, the residues showed moderate admixture between the wild and East Asian breeds (Supplementary Fig. 11). We then used the less-parameterized three- and four-population (F3/F4) test[38] to evaluate the statistical significance of these admixture events. Again, the F3 test strongly supported the admixture of dromedaries and Bactrian camels in KAZA, RUS, and IRAN (Supplementary Table 9). The more sensitive F4 test confirmed a significantly higher extent of admixture between dromedaries and Bactrian camels in Central Asia compared with those in East Asia (Supplementary Table 10). Among the latter, a higher extent of admixture with dromedaries was detected in XJ than in MG/IMG.

**Evidence for Central Asian origin by removing introgression.** East and Central Asia were the two alternative regions of domestication for Bactrian camels based on archaeological evidences[1,12,17], but the most probable one remained unsolved. Although we observed the largest genetic differentiation between the Iranian population and all the other domestic ones, the existence of admixture between dromedaries and Bactrian camels in Central Asian would weaken the support for origin inference. To reduce this effect, we attempted to remove the introgressed segments of dromedaries from the Bactrian camel genomes by using the "BABA/ABBA" test[39]. We grouped the East and Central Asian populations, and compared allele sharing between the two groups with dromedaries (Fig. 3a). As the ancestry of Bactrian camels in one dromedary, as well as the ancestry of domestic camels in three wild ones (Fig. 2b), would be confounding factors, we removed the four individuals in the analysis. We used the statistic $f_d$, a robust version of the Patterson's D, to locate introgressed segments[40] and applied a strict significance level of Z-score = 2 by using the Jackknife procedure (Supplementary Fig. 12). In a total of 21,153 non-overlapping 100 kb segments across the genome, there were far more segments with putative

signals of introgression in the Central Asian populations (11,711, Z-score > 2) than in the East Asian populations (3891, Z-score < −2) as expected. We then performed the Admixture analysis based on the remaining segments and confirmed that the introgression of dromedaries were effectively reduced (Supplementary Fig. 13). Re-calculation of the pairwise Fst after removing introgression still indicated that IRAN was the most differentiated one (0.04–0.06) among all the domestic populations (Fig. 3b). To gain more insights into the population phylogeny, we reconstructed the NJ tree based on the pairwise Fst and performed the bootstrap test (Fig. 3b). It confirmed that IRAN was the first one to separate among all the domestic Bactrian populations, followed by KAZA and RUS. The Bayesian binary Markov Chain Monte Carlo (MCMC) analysis based on the phylogeny strongly supported Central Asia as the ancestral area of domestic Bactrian camels (probability = 99.78%) and a subsequent dispersal route from Central to East Asia (Supplementary Fig. 14).

As an independent evidence, we also reconstructed the maximum likelihood tree of full-length mtDNAs based on the 128 samples we sequenced in this study, as well as 39 additional samples available from Genbank (Fig. 3c and Supplementary Table 11). Introgression of mtDNAs could easily be identified and excluded from the tree. For example, two newly sequenced mtDNAs from KAZA and RUS were clustered with dromedaries. Within the clade of domestic Bactrian camels, although most camels from different geographic regions were mixed, two mtDNAs from IRAN formed the most basal branches of the domestic populations (Fig. 3c). The Bayesian binary MCMC analysis again supported the Central Asian origin of domestic Bactrian camels (probability = 76.43%).

**Demographic history of Bactrian camels.** We performed several parametric modeling analyses to infer the demographic dynamics of the camels in history. Consistent with previous study[10], the long-term trajectories of Bactrian camels based on the pairwise sequentially Markovian coalescent (PSMC) model[41] revealed a tremendous decrease in the effective population size of the

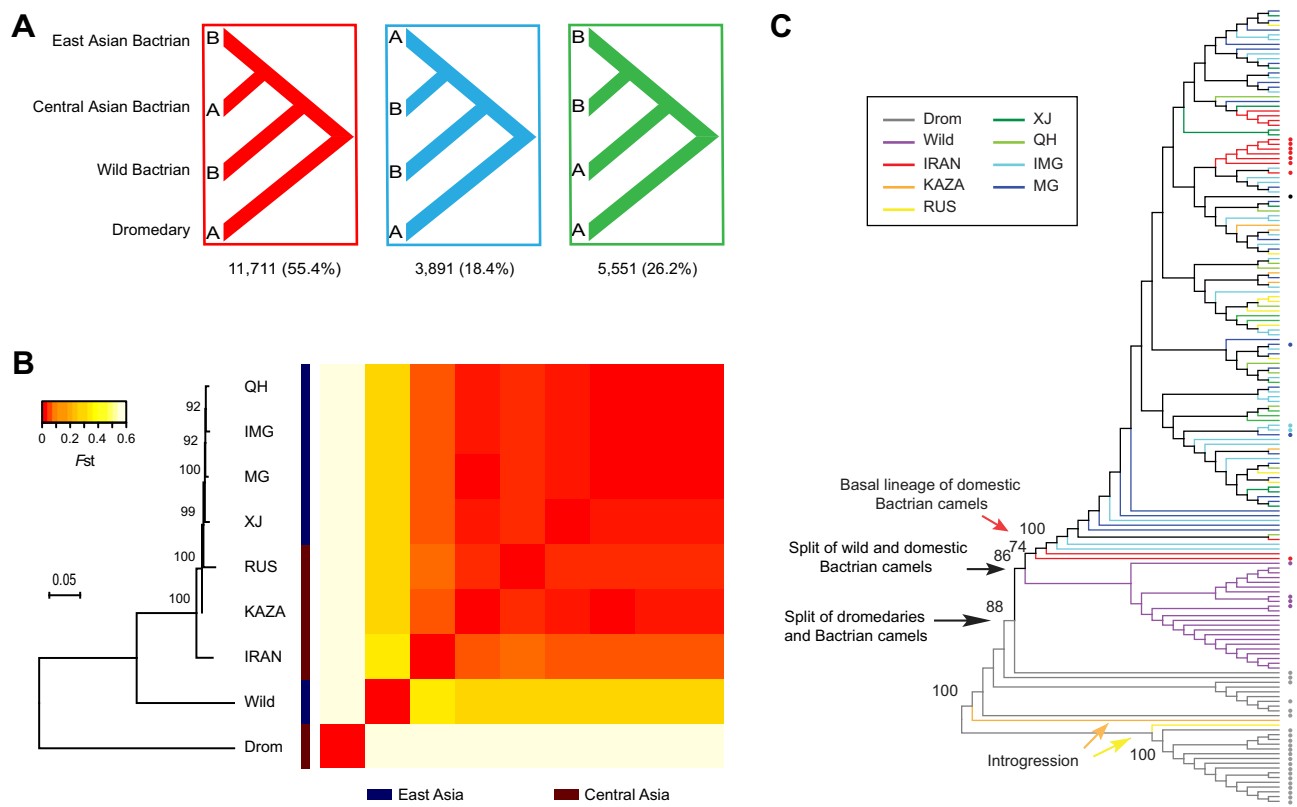

**Fig. 3 Identification of the origin of domestic Bactrian camels by removing introgression. a** BABA/ABBA analysis for introgression of dromedaries into domestic Bactrian camels. To focus on this introgression, one dromedary with the ancestry of Bactrian camels and three wild camels with the ancestry of domestic ones were removed. Number of 100 kb segments with significant $f_d$ (|Z-score| > 2) for each tree configuration is shown. **b** Neighbor-joining (NJ) tree of the populations after the introgressed segments were removed. The heatmap represents average pairwise $F$st for $5.1 \times 10^4$ 10 kb-sliding windows. Bootstrap values of the NJ tree were calculated by randomly sampling five thousand 10 kb windows for 100 times. **c** Maximum likelihood tree of full-length mtDNAs. Populations are represented by different colors and sequences from Genbank are indicated by dots. Bootstrap values for main branches are labeled.

ancestral camels earlier than one million years ago (Supplementary Fig. 15). Although the long-term trajectories of the wild and domestic Bactrian camels were generally similar, they were obvious to diverge from each other as early as 0.4 million years ago, excluding the former as direct progenitors of the latter as previous mtDNA analyses[9,14].

To explore the divergence time among the camel populations in more detail, we used the generalized phylogenetic coalescent sampler (G-PhoCS)[42]. Given the phylogeny of the camel populations, G-PhoCS could estimate the mutation-scaled population size and population divergence time based on unlinked neutral loci in individual genomes from each population. To reduce the model complexity, we only included dromedaries, wild Bactrian camels, and three representative populations (IRAN, KAZA, and MG) of domestic Bactrian camels (Supplementary Fig. 16 and Supplementary Table 12). According to Fig. 3b, IRAN and KAZA were the first two Central Asian populations to separate and the split of MG could indicate the dispersal from Central to East Asia. The age was calibrated by assuming the Bactrian camel and dromedary divergence of 5.73 million years according to the Timetree database[43]. When no migration band was incorporated, convergence of all parameter estimates could easily be achieved (Supplementary Fig. 17 and Supplementary Table 13). Similar to the PSMC results, the effective population size was generally decreased from ancestral to modern populations (Fig. 4). The divergence time between wild and domestic Bactrian camels was estimated as 0.43 million years

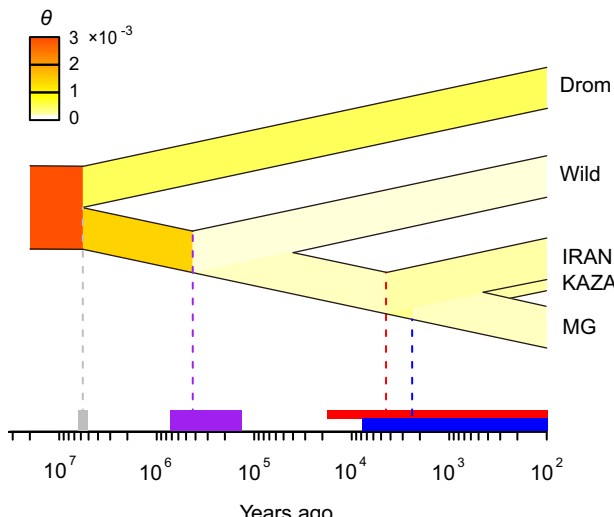

**Fig. 4 Parameter-based inference of demographic history with G-PhoCS.** The change in mutation-scaled effective population size $\theta$ is represented by heat colors. The time in years were calibrated by the divergence time between dromedaries and Bactrian camels. 95% Confidence intervals are shown by bars on the time axis. The red and blue bar indicate IRAN-MG and KAZA-MG divergence, respectively. These estimates are based on the model without migration.

ago (95% confidence interval [CI]: 0.13–0.73 Mya; Fig. 4), which was slightly later than that based on mtDNAs (0.7[14] or 1.1 Mya[9]). Among the domestic populations, IRAN was separated from others about 4.45 thousand years ago (95% CI: 0.07–17.6 Kya) and then the Central and East Asian populations were separated about 2.40 thousand years ago (95% CI: 0.01–7.84 Kya; Fig. 4).

To allow for gene flow, we also tried to introduce migration bands from dromedaries to Bactrian camel populations (Supplementary Fig. 16 and Supplementary Table 12). The estimates could only converge when a migration band from Iranian dromedaries to IRAN and a migration band from a ghost population to KAZA were introduced (Supplementary Fig. 18 and Supplementary Table 13). Although the divergence time between wild and domestic Bactrian camels was not changed with the migration model (0.46 Mya, 95% CI: 0.24–0.71 Mya), the first divergence time of the domestic populations (0.19 Mya, 95% CI: 0.08–0.31 Mya) became unrealistic, because it was far beyond the known history of livestock domestication (11.5 Kya[44]). Besides, the total migration rate was only estimated as 0.27% and 0.16% for the migration band to IRAN and KAZA, respectively, much lower than that estimated with Admixture (1–10%). A possible reason for the poor estimation would be a more complex admixture history than the continuous migration model with constant rates assumed by G-PhoCS.

## Discussion
In this study, we characterized for the first time the whole-genome variations of camels across Asia, including domestic Bactrian camels representing a major subset of recognized breeds, extant wild Bactrian camels as well as dromedaries. As the extant wild Bactrian camels are going towards extinction, our research provided extremely valuable genetic resources of the living fossil. Also, considering the extensive utilization of domestic camels in transportation, milk and wool production, our data provided new options to implement genetic association study and marker-assistant selection for improving livestock productivity and future breeding effects. In addition, these data provided an unprecedented opportunity to trace the origin and migration of domestic Bactrian camels in history.

Previous studies found limited archaeological records and molecular markers for the first domestication of Bactrian camels in Central Asia rather than in East Asia[17]. Here we provided more solid evidences on the basis of the whole-genome sequences. The earliest branching among the domestic Bactrian camels occurred between IRAN and all the others, which was followed by the split between the Central and East Asian populations. Although evident introgression of dromedaries was observed in Central Asia, we demonstrated that it will not influence our results by removing the introgressed genomic segments. In contrast, although the extant wild and domestic Bactrian camels share close habitat in MG, our whole-genome analyses gave a coherent result to other mtDNA analyses that the two populations were separated by so long a time that the latter were not likely to originate from the former[9,14]. Furthermore, the extant wild camels contributed little to the gene pool of domestic populations, implying that the domestic populations in MG could possibly be immigrated there during more recent periods.

Based on these results, we proposed a comprehensive scenario for the origin and migration of the Bactrian camels (Supplementary Fig. 19). After the ancestor of camels moved from North America and split into dromedaries and Bactrian camels, the wild Bactrian camels spread from East to Central Asia about 0.43 million years ago (95% CI: 0.13–0.73 Mya). The Bactrian camels were first domesticated in Central Asia before 4.45 thousand years ago (95% CI: 0.07–17.6 Kya), which were then migrated

back to East Asia around 2.40 thousand years ago (95% CI: 0.01–7.84 Kya) with the increasing economic exchange and cooperation between the West and East. This scenario could resolve the mystery why the wild and domestic Bactrian camels from the Mongolian Plateau have such a large genetic distance. It should be noted that the timing of the events here were based on the model without admixture. Considering that the domestic Bactrian camels in Central Asia were further hybridized with dromedaries out of Arabia, the origin and migration age of the domestic Bactrian camels would be overestimated because of the increased genomic divergence.

Despite the insights gleaned from our data, it was important to note that the direct wild progenitor of domestic Bactrian camels were not found in Central Asia now and may no longer exist. However, there were records suggesting that the wild Bactrian camels were more widely distributed throughout Asia in history, extending from the great bend of the Yellow River westward to central Kazakhstan[7]. In future work, sequencing of ancient genomes from camel fossils will add to the picture of their early domestication. Another issue in our study was that although the occurrence of gene flow between dromedaries and Bactrian camels in Central Asia was convincingly detected, the actual admixture history remained largely unknown. First, the TreeMix analysis suggested that although the Iranian Bactrian camels and dromedaries were directly mixed, those from KAZA and RUS appeared to be mixed with a ghost population (Fig. 2c). Second, when the excessive shared alleles between Iranian dromedaries and Central Asian Bactrian camels were removed, KAZA continued to have reduced divergence from the dromedaries compared with the other populations (population tree in Fig. 3b). This branching pattern was also consistent with the ghost admixture model for KAZA. Third, a continuous migration model with constant rates implemented by G-PhoCS could only capture a small fraction of admixture, even though a ghost population was assumed. These results hinted at a more complex and possible multistage admixture history between Bactrian camels and dromedaries. As we only had a few dromedaries from IRAN, a future attempt to collect dromedaries from more diverse populations could help to decipher the admixture history.

## Methods
**Sample preparation.** Blood samples of 105 domestic Bactrian camels were collected from villages in China (55), MG (28), KAZA (6), RUS (10), and IRAN (6). Blood samples of four dromedaries were also collected from IRAN. The collections were made during routine veterinary treatments with the guidelines from the Camel Protection Association of Inner Mongolia. An endeavor was made to collect samples from unrelated individuals based on the information provided by the owners and local farmers. We collected 50 ml blood for each camel from the jugular vein after disinfection treatment, placed it in EDTA anticoagulant tubes, and then stored it at −80 °C. Ear skin samples (0.5 cm) of 19 wild Bactrian camels were collected from the Great Gobi-Strictly Protected Area A in MG. The wild Bactrian camels chosen were artificially reared and the research was reviewed and approved by the Great Gobi National Park. Proper surgical procedures were adopted in the collection. Local anesthesia (5% procaine hydrochloride) was applied to the ear and the wound was disinfected with iodophor and sulfonamide powder. The samples were eluted with phosphate-buffered saline solutions, placed in cryotubes and were stored at −80 °C.

**Genome sequencing.** The genomic DNA was extracted from 200 μl blood samples with the QIAamp DNA Blood Mini kit (Qiagen) and from the skin samples with a standard phenol–chloroform method. The quality and integrity of DNA was controlled by OD260/280 ratio and agarose gel electrophoresis. For sequencing library preparation, the genomic DNA was sheared to fragments of 300–500 bp, which were then end repaired, 'A'-tailed, and ligated to Illumina sequencing adapters. The ligated products with sizes of 370–470 bp were selected on 2% agarose gels and then amplified by PCR. The libraries were sequenced on Illumina HiSeq platform with standard paired-end mode.

**Variant calling.** We used an in-house script to perform quality control on raw sequencing reads. Low-quality reads with ambiguous bases >10% were excluded.

The 3′-ends with base quality score <20 were trimmed and reads with length <35 bp were removed after trimming. Trimmed reads were mapped to the reference genome assembly of the Bactrian camel (ftp://ftp.ncbi.nih.gov/genomes/Camelus_ferus/CHR_Un/cfe_ref_CB1_chrUn.fa.gz) using BWA-MEM (v0.7.12)[45] for each individual and then processed with SAMtools (v1.3.1)[46]. We followed the GATK pipeline (v3.2–2)[47] for variant calling. First, PCR duplicates were removed using Picard tools (v1.135) and local indel realignment were performed. Second, SNPs and small indels were called with UnifiedGenotyper across all 128 individuals simultaneously. Finally, the raw variants were filtered with the following criteria: variant quality score >40, sequencing depth summing all individuals >200 and <5000, minor allele frequency >1%, variants with <20% individuals with missing genotypes, root mean square of mapping quality >30, and biallelic variants. Total number of SNPs were reduced from 17.76 to 13.83 million after filtering. Functional annotation of variants were performed with ANNOVAR (v2013-06-21)[48] according to RefSeq (ftp://ftp.ncbi.nih.gov/genomes/Camelus_ferus/GFF/ref_CB1_scaffolds.gff3.gz).

**Population statistics and structure**. Summary population statistics, including pairwise nucleotide diversity $\pi$, Watterson's $\theta$, and Weir's $F$st across 10 kb-sliding windows were calculated by VCFtools (v0.1.12b)[49]. Pairwise kinships between the samples were inferred by KING (v2.1.3)[50] and one of the paired individuals with close relationship was removed. For population structure analyses, SNPs in approximate linkage disequilibrium with each other were pruned by PLINK (v1.07)[51] (–indep-pairwise 50 5 0.5). SNPs located within exons and flanking 1 kb regions were also excluded. As a result, 2.08 million SNPs were preserved. MDS and pairwise distance matrix based on IBS were calculated using the –mds-plot 4 and –distance-matrix option in PLINK, respectively. The distance matrix was used to construct the NJ tree by Phylip (v3.69)[52]. One hundred random datasets were generated with –thin 0.1 option in PLINK and bootstrap values were retrieved from the consensus tree reconstructed by Phylip[52]. The population ancestry was inferred by ADMIXTURE (v1.3.0)[35] with a fast maximum likelihood method. The optimum number of ancestral clusters $K$ was estimated with the fivefold cross-validation procedure.

**TreeMix analysis and admixture tests**. Migration events among camel populations were inferred using TreeMix (v1.12)[37] with migration number $m = 0$–5. The threepop/fourpop module from the TreeMix package was used to perform the F3/F4 test[38,53] with -k 500. In the F3 test ($Z; X, Y$), one focal population ($Z$) was tested as a mixture of population $X$ and $Y$. A large negative value of F3 score (standardized to $Z$-score with the Jackknife procedure) would indicate a very strong signal of $Z$ as a mixture of $X$ and $Y$. In our analysis, we ran F3 tests with all configurations of the populations. In the more sensitive F4 test ($Y, Z; W, X$), where $W$ is an outgroup of $Y$ and $Z$, the admixture bias between $Y$ and $Z$ with $X$ was tested. If $Y$ (or $Z$) have more admixture with $X$, it will show significant negative (or positive) F4 score (standardized to $Z$-score with the Jackknife procedure). To focus on the admixture between the domestic Bactrian camels and dromedaries, we set the population configuration as ($Y, Z$; wild, drom), where $Y$ and $Z$ were two domestic populations.

**Local introgression test**. To select the local genomic regions with significant introgression between dromedaries and Bactrian camels after their divergence, we used an in-house script to perform the BABA/ABBA test[39] across 100 kb-sliding windows. For the tree configuration ($Y, Z; W, X$), the original Patterson's $D$ statistic can be calculated as a normalized F4 score[53]:

$$D = \frac{E((p_Y - p_Z)(p_W - p_X))}{E((p_Y + p_Z - 2p_Yp_Z)(p_W + p_X - 2p_Wp_X))} \quad (1)$$

where $p_X$ is the frequency of a given allele in population $X$ and the expectations $E()$ are estimated by averaging all SNPs in a window. The more robust $f_d$ statistic for local genomic regions, which is a special form of F4 ratio and directly measures the proportion of introgression[40], can be formulated as:

$$f_d = \begin{cases} \frac{E((p_Y - p_Z)(p_W - p_X))}{E(max((p_X - p_Y)(p_X - p_W), (p_Z - p_Y)(p_Z - p_W)))} & (D > 0) \\ \frac{E((p_Y - p_Z)(p_W - p_X))}{E(max((p_X - p_Z)(p_X - p_W), (p_Y - p_Z)(p_Y - p_W)))} & (D < 0) \end{cases} \quad (2)$$

We used the population configuration (East Asian, Central Asian; wild, drom) to perform the test. The $f_d$ statistic in each window was evaluated by the $Z$-score with the Jackknife procedure:

$$Z(f_d) = \frac{E(\widehat{f_d})}{\sqrt{var(\widehat{f_d}) \times n}} \quad (3)$$

where $\widehat{f_d}$ is estimated with a 10 kb block removed each time and $n$ is the repetition times.

**Population phylogeny and mtDNA analysis**. Population distance was measured with average $F$st across 10 kb windows. To minimize linkage disequilibrium and

perform the bootstrap test, five thousand 10 kb windows located at least 100 kb apart were randomly sampled for 100 times. The NJ tree and consensus tree were reconstructed by Phylip[52]. We complied the full-length mtDNA sequences of camels from those we sequenced in the study and those collected from GenBank. The sequences were aligned using ClustalW[54]. The control regions were deleted, because they were missing in many sequences and not well aligned. The maximum likelihood tree was constructed using MEGA (v6.06)[55] with 1000 random bootstrap runs. The Tamura-Nei model and uniform substitution rates among sites were adopted. The ancestral area inference was performed with the Bayesian binary MCMC method implemented in RASP (v4.0)[56]. The MCMC was run for 50,000 iterations with 100 iterations between two samples and the first 100 samples were discarded.

**G-PhoCS analysis**. To prepare for the G-PhoCS (v1.3)[42] input, we implemented the following filters to the genome to reduce the effects of selection and sequencing errors: exons and 1 kb flanking regions; gap regions in the genome assembly; and regions with repeat sequence annotations. Altogether, 47% of the genome were excluded. We then randomly collected ten thousand 1 kb loci located at least 30 kb apart, to ensure sufficient inter-locus recombination. Multiple sequence alignments for the loci from individual genomes per population were retrieved by vcf consensus in VCFtools[49], with heterozygous genotypes represented by the International Union of Pure and Applied Chemistry code and missing genotypes masked by "N." Recommended gamma priors were used in the G-PhoCS analysis for the mutation-scaled population size $\theta$, population divergence time $\tau$, as well as migration rate $m$. The MCMC was run for 100,000 burn-in iterations and 500,000 sampling iterations with 10 iterations between two traced samples. The automatic fine-tuning procedure was done during the first 10,000 iterations. The convergence and mixing of the MCMC trace were monitored by Tracer (v1.6, available from http://tree.bio.ed.ac.uk/software/tracer/). Because of the stochastic nature of the MCMC algorithm, we tested the models on independent datasets and accepted the results only if two independent runs achieved similar estimates. We explored four models as follows: no migration (model 1); a single migration band from the dromedary to IRAN (model 2); two migration bands from the dromedary to IRAN and KAZA, respectively (model 3); and a migration band from the dromedary to IRAN and another from a ghost population to KAZA (model 4). All loci of the ghost population were set as "N." Only the models 1 and 4 showed convergence within ten independent runs. The time scale in years was calibrated according to a consensus divergence time of Bactrian camels and dromedaries (5.73 Mya in TimeTree[43]). The total migration rate per band $M$ was calculated with $M = m\tau_m$, where $m$ was the mutation-scaled migration rate per generation and $\tau_m$ was the mutation-scaled time span of the migration band.

**Statistics and reproducibility**. Standard statistical tests were performed with R (v3.4.2). Specifically, the count of population-specific variants was compared between East Asia ($n = 4$) and Central Asia ($n = 3$) with the two-tailed $t$-test. The nucleotide diversity $\pi$ was compared between the populations with the two-tailed $t$-test based on twenty thousand 10 kb windows separated by 100 kb with each other. The ANOVA of missing count of variants was performed for domestic Bactrian camels ($n = 105$), wild Bactrian camels ($n = 19$), and dromedaries ($n = 4$), with sequencing depth as a covariate.

**Reporting summary**. Further information on research design is available in the Nature Research Reporting Summary linked to this article.

## Data availability

The raw data generated from this study have been submitted to the NCBI Sequence Read Archive (http://www.ncbi.nlm.nih.gov/sra/) under accession number SRP107089. The raw data are also available from NODE (http://www.biosino.org/node/) under accession number OEP000024. The datasets to reproduce the main figures have been submitted to Dryad (https://doi.org/10.5061/dryad.tx95x69sz).

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

## Acknowledgements

This work was supported by grants from the International S&T Cooperation Program of China (2015DFR30680 and ky201401002), the National Natural Science Foundation of China (31360397 and 31560710), the National Key R&D Program of China (2017YFA0505500, 2016YFC0901704, 2017YFC0907505, and 2017YFC0908405), the special project of the Inner Mongolia Autonomous Region, the Chinese Academy of Sciences (KFJ-STS-QYZD-126 and ZDBS-SSW-DQC-02), and the Youth Innovation Promotion Association CAS (2017325). We thank Dr Feng Qiu and Mr Ze Xu from BasePair Biotechnology Co., Ltd, for technical assistance on NGS. We thank Anhui Engineering Laboratory for Big Data of Precision Medicine, Suzhou, for providing computational resources.

## Author contributions

Y.L., Z.W., and J. designed research. S.H., G.C., T.J., N.H.-E., M.B., G.K.B., T.G.-E., B.T., W.Z., A.Z., H., E., A.N., P.M., N., G.M., C.N., O.K., Sirendalai, Sarentuya, and A. collected samples. L.M., L. Yi, G.D., J.H., and L.H. performed sequencing. L.M., L. Yuan, L. Yi, W.L., and Z.W. analyzed data. L.M., L. Yuan, Z.W., and J. wrote the manuscript.

## Competing interests

The authors declare no competing interests.

## Additional information

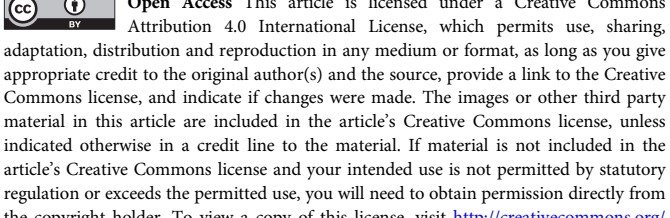

Liang Ming[1,2,25], Liyun Yuan[3,4,25], Li Yi[1,25], Guohui Ding[3,5], Surong Hasi[6], Gangliang Chen[7], Tuyatsetseg Jambl[8], Nemat Hedayat-Evright[9], Mijiddorj Batmunkh[10,11], Garyaeva Khongr Badmaevna[12], Tudeviin Gan-Erdene[13], Batsukh Ts[8], Wenbin Zhang[14], Azhati Zulipikaer[15], Hosblig[16], Erdemt[17], Arkady Natyrov[18], Prmanshayev Mamay[19], Narenbatu[20], Gendalai Meng[21], Choijilsuren Narangerel[22], Orgodol Khongorzul[1], Jing He[1], Le Hai[1], Weili Lin[3], Sirendalai[2], Sarentuya[2], Aiyisi[8], Yixue Li[3,4,23,24]⋆, Zhen Wang 🄳[3]⋆ & Jirimutu[1,2,10]⋆

[1]Key Laboratory of Dairy Biotechnology and Engineering, Ministry of Education, College of Food Science and Engineering, Inner Mongolia Agricultural University, Huhhot, China. [2]Inner Mongolia Institute of Camel Research, West Alax, Inner Mongolia, China. [3]Key Laboratory of Computational Biology, CAS-MPG Partner Institute for Computational Biology, Shanghai Institute of Nutrition and Health, Shanghai Institutes for Biological Sciences, Chinese Academy of Sciences, Shanghai, China. [4]Bio-Med Big Data Center, CAS-MPG Partner Institute for Computational Biology, Shanghai Institute of Nutrition and Health, Shanghai Institutes for Biological Sciences, Chinese Academy of Sciences, Shanghai, China. [5]Gui'an Bio-Med Big Data Center, Shanghai Institutes for Biological Sciences, Chinese Academy of Sciences, Guiyang, China. [6]Key Laboratory of Clinical Diagnosis and Treatment Technology in Animal Disease, Ministry of Agriculture, College of Veterinary Medicine, Inner Mongolia Agricultural University, Huhhot, China. [7]Bactrian Camel Academe of Altai, Xingjiang Wangyuan Camel Milk Limited Company, Fuhai County, Xijiang, China. [8]College of Industrial Technology, Mongolian University of Science and Technology, Ulaanbaater, Mongolia. [9]University of Mohaghegh Arabili, Ardabil, Iran. [10]China-Mongolia Joint Laboratory for Biomacromolecule Research, Ulaanbaatar, Mongolia. [11]Mongolian Wild Camel Protection Area, Ministry of Nature and Environment, Ulaanbaatar, Mongolia. [12]Kirovski Plant, Non-Public Joint-Stock Company, Republic of Kalmykia, Russia. [13]Institute of Chemistry and Chemical Technology, Mongolian Academy of Sciences, Ulaanbaatar, Mongolia. [14]Bactrian Camel Institute of Alxa, Inner Mongolia, China. [15]Animal Science Institute, Xinjiang Academy of Animal Science, Urumqi, China. [16]Animal Husbandry Bureau of North Urad, Bayannuur, Inner Mongolia, China. [17]Animal Husbandry Workstation of West Sunid, Xiliingol, Inner Mongolia, China. [18]Agrarian Faculty, Kalmyk State University, Republic of Kalmykia, Russia. [19]Kazakh National Agrarian University, Almaty, Kazakhstan. [20]College of Animal Science, Inner Mongolia Agricultural University, Huhhot, China. [21]Department of Pharmacy, Affiliated Hospital of Inner Mongolia Medical University, Huhhot, China. [22]Institute of Technology, Ulaanbaatar, Mongolia. [23]Shanghai Center for Bioinformation Technology, Shanghai Industrial Technology Institute, Shanghai, China. [24]Collaborative Innovation Center for Genetics and Development, Fudan University, Shanghai, China. [25]These authors contributed equally: Liang Ming, Liyun Yuan, and Li Yi. ⋆email: yxli@sibs.ac.cn; zwang01@sibs.ac.cn; yeluotuo1999@vip.163.com

