## [Peer Review File · Communications Biology]

Reviewers' comments:

Reviewer #1 (Remarks to the Author):

This paper, "Whole-genome sequencing of 128 camels across Asia provides insights into the origin and migration of domestic Bactrian camels", presents WGS data from a large number of domestic and wild Bactrian camels to investigate the history of the domestic Bactrian and its origin. In addition, several Dromedary camel genomes are presented to investigate admixture within the genus.

The key findings of the paper are:

Wild Bactrian camel form a distinct, divergent clade to domestic Bactrian, which appear to diverge from each other ~0.5 million years ago.

Within domestic Bactrian, Iranian domestics are an outgroup to all other domestics. This result is robust to Dromedary admixture (see below), suggesting a Central Asian domestication process. Bactrian from Central Asia appear to have low levels of Dromedary admixture; East Asian Bactrian do not.

Gene flow between wild and domestic Bactrian in East Asia is asymmetric, occurring from domestic to wild populations but not in the reverse direction.

In general the analyses presented in the paper were technically sound. For each question, the paper typically presents at least two separate, complementary analyses which are consistently in agreement. The sample sizes for the Bactrian are sufficiently large for accurate per-population estimates of genetic diversity. The authors wisely searched for and removed related individuals, which would have otherwise confounded diversity estimates. Admixture and Treemix analyses were performed according to widely-accepted standards (in some cases more stringent), with cross-validation error and residuals presented for each accordingly. For local admixture inference, the f_d statistic was employed, which is preferable than the D statistic, more commonly used for genome-wide admixture inference. The authors also tested several demographic models using G-PhoCS to estimate the population parameters of Bactrian camel domestication.

However, there were some issues with the alignment and variant calling steps which underpin most of the paper's findings; specifically, the choice of reference genome (domestic Bactrian) may have resulted in a bias toward reads with domestic Bactrian variants and against reads containing wild Bactrian/domestic Dromedary variants.

The questions addressed by this paper are in my opinion of broad interest and the paper represents an original and important contribution to the field of camel/domestication genetics. The paper is for the most part well structured, although some language changes are recommended for clarity and ease of reading. The analyses presented are comprehensive and the data set represents an extremely useful resource of the camelid genomics community. The authors are also to be commended for a thorough methods and supplementary section, and for employing a range of independent approaches to validate their results. However, the alignment issues mentioned above limits the strength of the paper's conclusions, and certain key point estimates (e.g. Bactrian camel domestication time of 10,000 yBP) are based off potentially-flawed data sets and analyses. I recommend that several key issues be addressed prior to publication, either by re-analyses using alternative data sets and filtering criteria, or by providing figures which would aid in assessing the potential issues highlighted below.

Major Comments

1. I have several notes of technical concern regarding their variant calling and filtering steps. According to the paper, filtering was performed on variants based on "average sequencing depth of individuals 1-40" and "individuals with missing genotypes <20" (lines 361-362). This minimum

sequencing depth is quite low e.g. see <https://doi.org/10.1038/s41598-018-38346-0> for a recommendation of >15X for accurate genotype calling. In addition "individuals with missing genotypes <20" is unclear - do the authors mean "sites with <20 individuals with missing genotypes"? This filtering decision could be mitigated by SNP quality control (which was performed), or an appropriate MAF filter (see below).

Later, variants are filtered to remove "SNPs with extremely low minor allele frequency (--maf 0.005)". By my calculations, this removes sites where only a single derived allele is observed (114 diploid unrelated individuals = 228 chromosomes; $228 * 0.005 = 1.14$ chromosomes). The combination of the very low mean coverage and MAF filter may have resulted in a substantial number of sequencing errors being included in the variant set, with consequences of downstream analyses e.g. individuals with higher error rates appearing more distant from other individuals. Linkage-disequilibrium filtering was also performed before MAF filtering, meaning that potential sequencing artifacts retained in the data may have affected variant correlation estimates and therefore LD filtering.

Given these filtering decisions, the authors should consider adding a supplementary figure illustrating the distribution of SNP quality score per individual (or something similar, such as rate of heterozygous calls per individual) following each filtering stage to demonstrate that the filtering strategy was sufficiently stringent. Ideally, a higher MAF filter would be used to ensure that variants observed in >1 individual are exclusively used (i.e. less likely to be sequencing error, and more useful for population-level analyses), but this may be problematic given the small number of Dromedary included in the analyses.

2. The decision to use a domestic Bactrian camel as the reference genome for alignment may have biased diversity estimates and other related statistics and methods e.g. Fst. As other papers have demonstrated that Dromedary are an outgroup to wild and domestic Bactrian camel, and wild Bactrian form a distinct clade with regards to domestic Bactrian, then both wild Bactrian and domestic Dromedary are more distantly related to the reference genome than domestic Bactrian are. This likely reduced the number of Dromedary/wild Bactrian reads containing true variants which aligned to the Bactrian reference genome, systematically biasing the variant set towards variants present in domestic Bactrian. As such, the diversity values presented e.g. Figure 1A, Sup Figure 4, Sup Table 3 are likely underestimates and/or do not accurately reflect the true patterns of shared diversity among these species. For example, Figure S4 shows that a very high proportion of Dromedary variants are also present in domestic Bactrian. This may be at least partially driven by the choice of reference genome biasing against Dromedary variants. Also, in Figure 1 Fst between domestic Bactrian and Dromedary/wild Bactrian may be inflated, due to the expected deficit of variant-containing reads aligning in the later, more divergent samples (as reduced diversity in one population increases Fst estimates).

Ideally, alignments would have been performed to an outgroup to the three sequenced species e.g. the Alpaca. However, this would necessitate a substantial amount of computational time. An alternative may be to present figures demonstrating the SNP missingness rate on an individual / population level. A high rate of missingness in the Dromedary and wild Bactrian genomes would give an indication that the choice of reference sequence was producing a systematic alignment issue for samples of these two species. Additionally, analyses which could be affected by the choice of reference genome could be so indicated in the text so readers can take this into consideration when assessing the findings.

3. The paper does not address the potential effect of their sample set being biased towards Chinese and Mongolian camel populations. Supplementary figures demonstrating heterozygosity rates per population and variant sites private to each population (after the final filtering step) would help in

assessing whether the set of variants used are particularly biased to those segregating in Chinese and Mongolian populations. If there is strong evidence of bias, sites could be filtered for those varying in multiple populations (e.g. an East Asian and non-East Asian population).

4. No mapping quality filter or filtering based on read mates properly pairing was performed (according to the text of the Methods, line 352-354). Sup Table S2 makes reference to "Properly mapped bases", but it is not indicated in the text whether proper pairing was used as a read filter. This should be clarified in the methods - was any mapping quality filter or mate-pairing filter performed prior to variant calling?

5. In the Discussion and Sup Fig S17, a model is proposed for the domestication history of Bactrian camels. However, the timing of several of the key events (e.g. Iranian-other domestics split) are estimated from a G-PhoCS model that does not include admixture (noting the authors attempts to fit such a model but with poor sample size of certain posterior estimates). Acknowledging that all models are wrong but some are useful, the parameter estimates from the "no migration" model are presented without any highlighting that including migration would reduce the age of those events i.e. artificially increased divergence of Iranian Bactrian from other bactrian. This seems quite important given that the paper has convincingly demonstrated the occurrence of gene flow from Dromedary to Bactrian camel. The text should acknowledge the issues with these estimates, preferably using credible ranges rather than point estimates in their proposed model of Bactrian camel domestication in the Discussion (lines 306-317) and elsewhere.

6. Similarly, the model presented in Sup Fig S17 explicitly suggests Iran as the domestication region of Bactrian camel. This does not account for other, unsampled, modern Bactrian camel populations. Although this paper presents a dramatic increase in Bactrian camel genomic resources, its sampling area is still biased substantially towards East Asia. Other Central Asian populations may be better proxies for the original source of Bactrian camel. The authors are careful in the text to suggest a Central Asian domestication rather than Iranian, but this is not the case in Fig S17 - the figure and legend be amended to account for this.

7. The approach taken in the section starting at line 217 and illustrated in Figure 3A e.g. to remove regions of the genome which show ancestral alleles in Central Asian Bactrian and Dromedaries, but derived alleles in East Asian and Wild - is an interesting one. However, a flaw in the implementation taken in the paper is the absence of a true outgroup; just as Iranian Bactrian are admixed with Dromedary, the Dromedary sequenced here show ancestry related to Iranian Bactrian. Of the three Dromedary genomes presented, one shows the Blue ($K=3$) Bactrian-like ancestry; this individual could be removed from the D and fd calculations. Alternatively and preferably, this analysis could be strengthened by inclusion of a true outgroup genome e.g. an alpaca or llama, so regions which Dromedary and Central Asians, but not East Asians, share the derived allele, could be identified directly.

8. The method illustrated in Figure 3A, in which regions which show an excess of Central Asian Bactrian and Dromedarius allele sharing are removed, will also remove regions in which East Asian Bactrian and Wild Bactrian share an excess of alleles. However, in Sup Figure 11 Wild Bactrian continue to have "East Asian like" ancestry (blue component). The authors could indicate if that component has decreased between Admixture runs (Figure 2B and Figure S11), and their hypotheses if it has not - perhaps wild-domestic admixture was an old event, between a domestic population not well represented by modern East Asian domestics (as suggested by lines 303-304)?

9. According to Figure 3B, Kazak Bactrian continue to have a reduced divergence from Dromedary compared to other domestics; the approach illustrated in Figure 3A does not appear to have removed

the traces of admixture in this population. This supports the KAZA population having traces of admixture with a ghost population, as suggested by Figure S14, and hints at a more complex, multistage admixture history between domestic Bactrian and Dromedary. This observation should be highlighted in the main text or discussion.

Minor Comments

1. The abstract lacks details on “what question remains” for Bactrian camel domestication. The abstract also states “confirming they are a separate species”, but this is not discussed in the results section. That dromedary and bactrian camels are evidenced to be distinct species is no longer an issue of contention and I recommend that the “separate species” sentence be deleted.

2. Introduction: broader sentences (eg line 89: “For example, the Bactrian..”) should have a citation. The importance of Bactrian camel to livelihoods of many worldwide today could also be mentioned.

3. The introduction should cite recent Y chromosome evidence showing strong divergence of domestic and wild Bactrian, with some evidence of admixture (Felkel et al 2019).

4. The methods section should indicate which GATK calling algorithm was used (i.e. UnifiedGenotyper).

5. The estimates of m (migration band) in Sup Table S13 - it should be clear to the reader if m refers to numbers of individuals, or a rate per generation, or per band. If per band, what is the generation rate per band?

6. The main text at line 278 indicates that the total migration rate was “1% of both migration bands”, but it is unclear how 1% relates to the scaled migration values in Table S13. The methods at line 446 indicate that the migration rate was calculated by $M = \pi m$, but the values for πm are not clearly indicated in the tables S12 and S13, or the methods section.

7. The authors should clarify (in the Supplementary section beginning at line 425) why RUS was not included in the G-PhoCS modelling.

8. The authors should be more explicit in their methodology of exploring model space for the G-PhoCS analyses with migration. For example, it is not clear if a ghost migration from the Dromedary clade to IRAN was tested.

9. In Figure 4, why is KAZ not presented? The split time estimate is indicated, but the KAZA-MG node and KAZA branch are not illustrated.

10. Figure 4 could benefit from a more divergent colour scale (e.g. red to white not red to yellow).

11. The Figure 4 legend should clearly indicate that these estimates are from the NO-MIG model.

12. At line 352-353, when the authors state “Low-quality reads marked by Illumina sequencers in the FASTQ files were removed”, what exactly are they referring to? The sentence refers to trimming low quality bases and removing short sequences, but not “low-quality reads”. What are the criteria for “low quality reads”?

13. Line 316-317 - the comparison to the expansion of modern non-Africans into Eurasia may not be an appropriate one, given the admixture known to occur between indigenous Eurasian (e.g.

Neanderthal). The reference to "a small group of people out of Africa" is also confusing, given the population increase that likely occurred during the Out-of-Africa expansion. I recommend the lines from 314-317 referring to modern human evolution be removed.

14. The text of the document requires careful editing for grammar and use of expressions e.g. line 63, "made also little contribution" to "contributed little"; line 69, "where the native wild" to "where native wild"; line 77 "latter herds wildly line" to "latter wild herds live"; line 313, "could well resolve the mystery" to "could resolve the mystery"; line 320 "which may no longer exist" to "and may no longer exist"; line 362 "individuals with missing genotypes <20" to "genotypes with <20 missing individuals"

Reviewer #2 (Remarks to the Author):

Ming and colleagues sequenced 128 camels to understand the complex evolutionary history of Bactrian camels, since their divergence from dromedaries and the endangered wild Bactrian camel. I already revised this manuscript a few months ago, for another journal of the Nature group, and I find this shortened version more clear and concise. Authors have implemented most of my comments since then. Therefore, I find their work suitable for publication, pending some minor suggestions and questions that remain unclear to me:

- in the abstract, the authors state that "The domestic Bactrian camels were treated as the principal means of locomotion between the eastern and western cultures in history." This might be true for the the Silk Road, but it cannot be generalised beyond this. Horse domestication was for example more crucial in enhancing human mobility. See Loog et al. 2017 PNAS for a reconstruction of the human migration rates through time, where authors highlight the impact of horse riding. Also multiple studies in ancient DNA showing how the horse herders and riders, such as the Yamnaya people, changed the Euroasiatic genomic landscape during the Iron Age.

- Studies related to horse domestication should be cited, at least regarding the observed pattern of less diversity in wild than in domestic camels. The authors claim this different from pigs and dogs. But horses show the same: Wild (or feral) Przewalski's are less diverse than their clearly domestic counterparts. It is interesting because both species increased human mobility.

- the ti/tv ratio appears to be quite high in Figure S3 and Table S3. Do the authors have any explanation for this? These ti/tv statistics seem to be prior to SNP quality filtering, otherwise they would not have ~18 million SNP (Figure S3), but 15.76 millions. I suspect the ti/tv increases a bit more after filtering (i.e. false positives are often tv). In my opinion, authors should clarify the reasons underlying their ti/tv. As a reader, it is suspiciously intriguing. One possible explanation for this elevated ti/tv pertains to the join GATK calling including individuals from multiple populations/species. Please explore and clarify.

- the residuals in treemix are insanely high (Figure S9). Authors highlight TreeMix reconstruction with $m=3$, but $m = 4$ actually seems to produce significantly lower residuals. Why not using $m = 4$ in the main then? Is it because the 4th admixture event involves Drom->XJ_TLM (Table S8), with XJ_TLM being a Chinese breed not from Central Asia? Please clarify. An objective criteria is expected from a reader perspective.

- unless I am miss-interpreting it, I cannot really see that Fig S11 shows that introgressed tracts have been removed from the camel genome after the f_d analysis. For $K=5$, it is evident that there is some red (IRAN component) within Dromedaries. It might be just a problem with plotting.

- Based on Fig S13, the N_e trajectories of wild and domestic camels seem to diverge 0.4 or 0.3 mya, instead of the 0.5 mya stated in the main.

- For Fig S14, please add the coverage of the samples used for the G-PhoCS analysis. Only high coverage samples can be used for this analysis (and probably for PSMC). With only moderate depth (8-15x), many heterozygous sites remain undiscovered. This is essential for analyses relying on a single genome per population.

- In Fig S17, authors draw "domestication in Iran". I think they should say "domestication in Central Asia" instead, as they do in the main. They lack samples from other regions in Central Asia to claim that domestication occurred specifically in Iran.

- In the main, the migration bands inferred by G-PhoCS are reported as 1%. However, in table S13, it appears to be higher (2.225% and 2.170%). Please clarify.

- In the same table s13, the model with migration predicts that the split of dromedaries and wild animals occurred at the same time (35,300 ya)? Please justify this, as it could reflect a model mis-specification.

- Authors claim that "Although dromedaries were more divergent from both of the Bactrian camel species in phylogeny, the domestic Bactrian camels shared more variants with the dromedaries (52.69%) than with the wild Bactrian camel (31.14%) (Supplementary Figure S4) due to the tremendous reduction of genetic variants observed in the extant wild Bactrian camel.". Given their results, I think authors should add that: "... due to the tremendous reduction of genetic variants observed in the extant wild Bactrian camel and to gene flow between dromedaries and domestic Bactrian camels." (or something similar)

- ADMIXTURE analysis reveals a bit of blue component (domestic Bactrian camels) in wild Bactrian camels. It appears to be significant at least for a few individuals. Based on TreeMix residuals, the donor population could be MG_TH. Authors should discuss about this possibility. F4 statistics are compatible with this. This is important given the endangered status of wild camels (perhaps not so wild then). The authors disregard this possibility and state that wild Bactrian camels have not exchanged genes with domestic livestock.

- The authors state that "The differentiation among the domestic Bactrian camels was much lower, implying a single-origin of them.". Despite probably true, I feel the verb "implying" is too strong here. I do not think we can infer a single origin based on pairwise F_{st} values. The tree, where domestic Bactrian camels are monophyletic, is more conclusive. Please re-phrase to something as: "The differentiation among the domestic Bactrian camels was much lower, in line with their recent single origin".

Hope you find these comments positive and constructive, because it is an interesting work (as stated in my previous review), and only requires some minor clarifications. My name is Pablo Librado.

Responses to Reviewer #1

We appreciate Reviewer #1's constructive review and comments, and we have revised the manuscript taking the suggestions into account. Specifically, we reformed the SNP filtering with more stringent criteria and updated the results with the new dataset. We also evaluated the reference bias with synonymous SNPs as suggested by the Editor. Please find our point-to-point responses to the comments below.

Major Comments

1. I have several notes of technical concern regarding their variant calling and filtering steps. According to the paper, filtering was performed on variants based on "average sequencing depth of individuals 1-40" and "individuals with missing genotypes <20" (lines 361-362). This minimum sequencing depth is quite low e.g. see <https://doi.org/10.1038/s41598-018-38346-0> for a recommendation of >15X for accurate genotype calling. In addition "individuals with missing genotypes <20" is unclear - do the authors mean "sites with <20 individuals with missing genotypes"? This filtering decision could be mitigated by SNP quality control (which was performed), or an appropriate MAF filter (see below).

Response: We are sorry for the confusion because variant filtering based on individuals was misleading. As a population-level analyses, sequencing depth summing all individuals should be used. In the revised manuscript, a minimum depth of 200X was used, which was set to identify variants with MAF = 1% (see below) with at least two alternative alleles. We also clarify the criteria "variants with <20% individuals with missing genotypes". The filtering steps were described at Line 407-410, and the SNP quality metrics following each filtering step were illustrated in Supplementary Fig. S3.

Later, variants are filtered to remove "SNPs with extremely low minor allele frequency (--maf 0.005)". By my calculations, this removes sites where only a single derived

*allele is observed (114 diploid unrelated individuals = 228 chromosomes; $228 * 0.005 = 1.14$ chromosomes). The combination of the very low mean coverage and MAF filter may have resulted in a substantial number of sequencing errors being included in the variant set, with consequences of downstream analyses e.g. individuals with higher error rates appearing more distant from other individuals. Linkage-disequilibrium filtering was also performed before MAF filtering, meaning that potential sequencing artifacts retained in the data may have affected variant correlation estimates and therefore LD filtering.*

Given these filtering decisions, the authors should consider adding a supplementary figure illustrating the distribution of SNP quality score per individual (or something similar, such as rate of heterozygous calls per individual) following each filtering stage to demonstrate that the filtering strategy was sufficiently stringent. Ideally, a higher MAF filter would be used to ensure that variants observed in >1 individual are exclusively used (i.e. less likely to be sequencing error, and more useful for population-level analyses)), but this may be problematic given the small number of Dromedary included in the analyses.

Response: Following the reviewer's calculation, we increased the MAF to 1%, which ensure that variants observed in >1 individual are used. Supplementary Fig. S3 illustrated the distribution of SNP quality per individual. It was shown that with this MAF filter, variants private to each individual were completely eliminated.

2. The decision to use a domestic Bactrian camel as the reference genome for alignment may have biased diversity estimates and other related statistics and methods e.g. F_{st} . As other papers have demonstrated that Dromedary are an outgroup to wild and domestic Bactrian camel, and wild Bactrian form a distinct clade with regards to domestic Bactrian, then both wild Bactrian and domestic Dromedary are more distantly related to the reference genome than domestic Bactrian are. This likely reduced the number of Dromedary/wild Bactrian reads containing true variants which aligned to the Bactrian reference genome, systematically biasing the variant set

towards variants present in domestic Bactrian. As such, the diversity values presented e.g. Figure 1A, Sup Figure 4, Sup Table 3 are likely underestimates and/or do not accurately reflect the true patterns of shared diversity among these species. For example, Figure S4 shows that a very high proportion of Dromedary variants are also present in domestic Bactrian. This may be at least partially driven by the choice of reference genome biasing against Dromedary variants. Also, in Figure 1 Fst between domestic Bactrian and Dromedary/wild Bactrian may be inflated, due to the expected deficit of variant-containing reads aligning in the later, more divergent samples (as reduced diversity in one population increases Fst estimates).

Ideally, alignments would have been performed to an outgroup to the three sequenced species e.g. the Alpaca. However, this would necessitate a substantial amount of computational time. An alternative may be to present figures demonstrating the SNP missingness rate on an individual / population level. A high rate of missingness in the Dromedary and wild Bactrian genomes would give an indication that the choice of reference sequence was producing a systematic alignment issue for samples of these two species. Additionally, analyses which could be affected by the choice of reference genome could be so indicated in the text so readers can take this into consideration when assessing the findings.

Response: Regarding the reference bias, we first compared the missing rate among the three species. Because the missing rate was also correlated with sequencing depth, we performed analysis of variance (ANOVA), taking the sequencing depth as a covariate (Supplementary Table S7). As a result, there was no significant difference in the missing rate between the domestic and wild Bactrian camels ($P = 0.50$), but there was significant difference between the Bactrian camels and dromedaries ($P = 0.004$), suggesting existence of the reference bias between the latter two species. Considering the divergence time of the species (wild-domestic Bactrian camel: ~ 0.4 Mya, Bactrian camel-dromedary: ~ 5 Mya, camel-alpaca: ~ 15 Mya), using an outgroup as reference such as alpaca would also be problematic because the missing rate would be too high. To further evaluate effect of the reference bias to our estimates, we adopted the

Editor's suggestion to estimate diversity and F_{st} with just synonymous SNPs because reads in exons are more likely to map to a slightly more distant reference genome. Supplementary Fig. S7 showed that the estimate based on whole genomes were in good linearity with that based on synonymous SNPs across all species. So we think the effect of reference bias to our estimates was minor. We indicated the reference bias and added the analyses at Line 183-193.

3. The paper does not address the potential effect of their sample set being biased towards Chinese and Mongolian camel populations. Supplementary figures demonstrating heterozygosity rates per population and variant sites private to each population (after the final filtering step) would help in assessing whether the set of variants used are particularly biased to those segregating in Chinese and Mongolian populations. If there is strong evidence of bias, sites could be filtered for those varying in multiple populations (e.g. an East Asian and non-East Asian population).

Response: We summarized the variants for East and Central Asian populations (Line 149-153). In total, there were 12.68 million and 11.61 million variants in the two groups of populations, respectively (Supplementary Fig. S4). There was no significant difference in the private variants to each population between the two groups ($P = 0.77$, Supplementary Table S4). This was because although the sample size was biased towards East Asian populations, the heterozygosity was higher in Central Asian populations.

4. No mapping quality filter or filtering based on read mates properly pairing was performed (according to the text of the Methods, line 352-354). Sup Table S2 makes reference to "Properly mapped bases", but it is not indicated in the text whether proper pairing was used as a read filter. This should be clarified in the methods - was any mapping quality filter or mate-pairing filter performed prior to variant calling?

Response: We added a mapping quality filter, i.e. root mean square of mapping

quality >30 (Line 410). We did not use proper pairing as a read filter because it would exclude genomic regions that showed structural difference with the draft reference genome.

5. In the Discussion and Sup Fig S17, a model is proposed for the domestication history of Bactrian camels. However, the timing of several of the key events (e.g. Iranian-other domestics split) are estimated from a G-PhoCS model that does not include admixture (noting the authors attempts to fit such a model but with poor sample size of certain posterior estimates). Acknowledging that all models are wrong but some are useful, the parameter estimates from the “no migration” model are presented without any highlighting that including migration would reduce the age of those events i.e. artificially increased divergence of Iranian Bactrian from other bactrian. This seems quite important given that the paper has convincingly demonstrated the occurrence of gene flow from Dromedary to Bactrian camel. The text should acknowledge the issues with these estimates, preferably using credible ranges rather than point estimates in their proposed model of Bactrian camel domestication in the Discussion (lines 306-317) and elsewhere.

Response: We discussed the issue that the ages based on the “no migration” model would be overestimated (Line 349-352). We also added credible ranges for the estimates throughout the manuscript (Line 344-346, Supplementary Fig. S19).

6. Similarly, the model presented in Sup Fig S17 explicitly suggests Iran as the domestication region of Bactrian camel. This does not account for other, unsampled, modern Bactrian camel populations. Although this paper presents a dramatic increase in Bactrian camel genomic resources, its sampling area is still biased substantially towards East Asia. Other Central Asian populations may be better proxies for the original source of Bactrian camel. The authors are careful in the text to suggest a Central Asian domestication rather than Iranian, but this is not the case in Fig S17 - the figure and legend be amended to account for this.

Response: We agree that we can only conclude a Central Asian rather than Iranian origin. We corrected the legend in Supplementary Fig. S19.

7. The approach taken in the section starting at line 217 and illustrated in Figure 3A e.g. to remove regions of the genome which show ancestral alleles in Central Asian Bactrian and Dromedaries, but derived alleles in East Asian and Wild - is an interesting one. However, a flaw in the implementation taken in the paper is the absence of a true outgroup; just as Iranian Bactrian are admixed with Dromedary, the Dromedary sequenced here show ancestry related to Iranian Bactrian. Of the three Dromedary genomes presented, one shows the Blue (K=3) Bactrian-like ancestry; this individual could be removed from the D and fd calculations. Alternatively and preferably, this analysis could be strengthened by inclusion of a true outgroup genome e.g. an alpaca or llama, so regions which Dromedary and Central Asians, but not East Asians, share the derived allele, could be identified directly.

Response: We agree that the ancestry of Bactrian camels in dromedaries would confound the ancestry of dromedaries in Bactrian camels with the ‘BABA/ABBA’ test. We followed the reviewer’s first suggestion to remove the dromedary that shows Bactrian-like ancestry (Line 248-250, Supplementary Fig. S13).

8. The method illustrated in Figure 3A, in which regions which show an excess of Central Asian Bactrian and Dromedarius allele sharing are removed, will also remove regions in which East Asian Bactrian and Wild Bactrian share an excess of alleles. However, in Sup Figure 11 Wild Bactrian continue to have “East Asian like” ancestry (blue component). The authors could indicate if that component has decreased between Admixture runs (Figure 2B and Figure S11), and their hypotheses if it has not - perhaps wild-domestic admixture was an old event, between a domestic population not well represented by modern East Asian domestics (as suggested by lines 303-304)?

Response: The East Asian ancestry in wild camels could be decreased with our method. However, just like the reviewer's last comment, the admixture between the East Asian and wild camels would confound the admixture between the Central Asian camels and dromedaries. So we think a better way is to remove the three wild camels that show East Asian ancestry in the 'BABA/ABBA' test (Line 248-250, Supplementary Fig. S13).

9. According to Figure 3B, Kazak Bactrian continue to have a reduced divergence from Dromedary compared to other domestics; the approach illustrated in Figure 3A does not appear to have removed the traces of admixture in this population. This supports the KAZA population having traces of admixture with a ghost population, as suggested by Figure S14, and hints at a more complex, multistage admixture history between domestic Bactrian and Dromedary. This observation should be highlighted in the main text or discussion.

Response: This is an interesting observation and we discussed the possibility of a ghost population at Line 363-367. Meanwhile, we highlighted that a more complex and multistage admixture history might exist between Bactrian camels and dromedaries. We think this point of view could also help to explain why we could only get poor estimates with a simple migration model assumed by G-PhoCS (Line 367-370).

Minor Comments

1. The abstract lacks details on "what question remains" for Bactrian camel domestication. The abstract also states "confirming they are a separate species", but this is not discussed in the results section. That dromedary and bactrian camels are evidenced to be distinct species is no longer an issue of contention and I recommend that the "separate species" sentence be deleted.

Response: In Abstract, we added “whether they originated from East Asia or Central Asia remains elusive” (Line 58-59). We also deleted “confirming they are separate species”.

2. Introduction: broader sentences (eg line 89: "For example, the Bactrian..") should have a citation. The importance of Bactrian camel to livelihoods of many worldwide today could also be mentioned.

Response: We added references for the sentences and mentioned the importance of Bactrian camels today (Line 91-92).

3. The introduction should cite recent Y chromosome evidence showing strong divergence of domestic and wild Bactrian, with some evidence of admixture (Felkel et al 2019).

Response: The recent study of Y chromosomes was cited (Line 101, Line 213).

4. The methods section should indicate which GATK calling algorithm was used (i.e. UnifiedGenotyper).

Response: GATK UnifiedGenotyper was mentioned (Line 407).

5. The estimates of m (migration band) in Sup Table S13 - it should be clear to the reader if m refers to numbers of individuals, or a rate per generation, or per band. If per band, what is the generation rate per band?

Response: In Supplementary Table S13, ‘m’ is the mutation-scaled migration rate per generation. We clarified the relationship between ‘m’ and ‘M’ (total migration rate per band) with footnotes of the table.

6. *The main text at line 278 indicates that the total migration rate was “1% of both migration bands”, but it is unclear how 1% relates to the scaled migration values in Table S13. The methods at line 446 indicate that the migration rate was calculated by $M = m\tau m$, but the values for τm are not clearly indicated in the tables S12 and S13, or the methods section.*

Response: As the last comment, the conversion between ‘m’ and ‘M’ was illustrated in Supplementary Table S13 and explained at Line 493-495.

7. *The authors should clarify (in the Supplementary section beginning at line 425) why RUS was not included in the G-PhoCS modelling.*

Response: To reduce model complexity, we only chose one Central Asian population (KAZA) and one East Asian population (MG) besides IRAN. KAZA was chosen because it was the first one to separate next to IRAN (Fig. 3B). We explained this at Line 289-293.

8. *The authors should be more explicit in their methodology of exploring model space for the G-PhoCS analyses with migration. For example, it is not clear if a ghost migration from the Dromedary clade to IRAN was tested.*

Response: We described the four models we explored at Line 487-490, i.e. 1) no migration; 2) dromedary to IRAN; 3) dromedary to IRAN and dromedary to KAZA; 4) dromedary to IRAN and ghost to KAZA. Only the model 1) and 4) showed convergence results.

9. *In Figure 4, why is KAZ not presented? The split time estimate is indicated, but the KAZA-MG node and KAZA branch are not illustrated.*

Response: The KAZA branch is partially hidden by IRAN. We added border lines in

the figure to show them.

10. Figure 4 could benefit from a more divergent colour scale (e.g. red to white not red to yellow).

Response: We used the color scale from red to white as suggested.

11. The Figure 4 legend should clearly indicate that these estimates are from the NO-MIG model.

Response: We clarified that the estimates were based on the model without admixture in the legend.

12. At line 352-353, when the authors state “Low-quality reads marked by Illumina sequencers in the FASTQ files were removed”, what exactly are they referring to? The sentence refers to trimming low quality bases and removing short sequences, but not “low-quality reads”. What are the criteria for “low quality reads”?

Response: We excluded low-quality reads with N% > 10% (clarified at Line 400).

13. Line 316-317 - the comparison to the expansion of modern non-Africans into Eurasia may not be an appropriate one, given the admixture known to occur between indigenous Eurasian (e.g. Neanderthal). The reference to “a small group of people out of Africa” is also confusing, given the population increase that likely occurred during the Out-of-Africa expansion. I recommend the lines from 314-317 referring to modern human evolution be removed.

Response: We deleted the comparison with human evolution in Discussion.

14. The text of the document requires careful editing for grammar and use of

expressions e.g. line 63, “made also little contribution” to “contributed little”; line 69, “where the native wild” to “where native wild”; line 77 “latter herds wildly line” to “latter wild herds live”; line 313, “could well resolve the mystery” to “could resolve the mystery”; line 320 “which may no longer exist” to “and may no longer exist”; line 362 “individuals with missing genotypes <20” to “genotypes with <20 missing individuals” .

Response: We checked the grammar carefully and those mentioned by the reviewers were corrected.

Responses to Reviewer #2

We appreciate Dr. Pablo Librado's constructive review and comments, and we have revised the manuscript taking the suggestions into account. Please find our point-to-point responses to the comments below.

- in the abstract, the authors state that "The domestic Bactrian camels were treated as the principal means of locomotion between the eastern and western cultures in history." This might be true for the the Silk Road, but it cannot be generalised beyond this. Horse domestication was for example more crucial in enhancing human mobility. See Loog et al. 2017 PNAS for a reconstruction of the human migration rates through time, where authors highlight the impact of horse riding. Also multiple studies in ancient DNA showing how the horse herders and riders, such as the Yamnaya people, changed the Euroasiatic genomic landscape during the Iron Age.

Response: We agree with the comment. In Abstract we indicated that the camels were one of the principal means of locomotion (Line 57). We also mentioned the role of horses in promoting human mobility at Line 88.

- Studies related to horse domestication should be cited, at least regarding the observed pattern of less diversity in wild than in domestic camels. The authors claim this different from pigs and dogs. But horses show the same: Wild (or feral) Przewalski's are less diverse than their clearly domestic counterparts. It is interesting because both species increased human mobility.

Response: It is interesting that both wild camels and horses are less diverse than their domestic counterparts, probably because both wild species are endangered. We cited the study of the Przewalski's horse (Sarkissian, et al. Curr Biol 25, 2577) at Line 167.

- the ti/tv ratio appears to be quite high in Figure S3 and Table S3. Do the authors have

any explanation for this? These ti/tv statistics seem to be prior to SNP quality filtering, otherwise they would not have ~18 million SNP (Figure S3), but 15.76 millions. I suspect the ti/tv increases a bit more after filtering (i.e. false positives are often tv). In my opinion, authors should clarify the reasons underlying their ti/tv. As a reader, it is suspiciously intriguing. One possible explanation for this elevated ti/tv pertains to the joint GATK calling including individuals from multiple populations/species. Please explore and clarify.

Response: The ti/tv ratio is specific to each species. We noted that a recent study also reported a higher ti/tv ratio (2.31-2.34) in dromedaries than other species (Khalkhali-Evrigh, et al. Plos One 13, doi:10.1371/journal.pone.0204028). In our dataset, the ti/tv ratio was 2.29 prior to SNP quality filtering, and it was increased to 2.44 following the filtering procedures. So the high ti/tv was a result of the filtering stringency. We explained the elevated ti/tv ratio at Line 138-141, and illustrated the change of ti/tv ratio following each filtering stage in Supplementary Fig. 3.

- the residuals in treemix are insanely high (Figure S9). Authors highlight TreeMix reconstruction with $m=3$, but $m = 4$ actually seems to produce significantly lower residuals. Why not using $m = 4$ in the main then? Is it because the 4th admixture event involves Drom->XJ_TLM (Table S8), with XJ_TLM being a Chinese breed not from Central Asia? Please clarify. An objective criteria is expected from a reader perspective.

Response: $m = 4$ indeed continued to decrease the residues and we used $m = 4$ in Figure 2C. The admixture between the dromedary and a XJ breed is not surprising because XJ is a bridge across Central and East Asia.

- unless I am miss-interpreting it, I cannot really see that Fig S11 shows that introgressed tracts have been removed from the camel genome after the f_d analysis. For $K=5$, it is evident that there is some red (IRAN component) within Dromedaries. It might be just a problem with plotting.

Response: The ancestry of Bactrian camels in dromedaries would confound the ancestry of dromedaries in Bactrian camels with the ‘BABA/ABBA’ analysis. So we followed reviewer 1’s suggestion to remove the dromedary that shows Bactrian-like ancestry before the analysis (Line 248-250, Supplementary Fig. S13).

- Based on Fig S13, the Ne trajectories of wild and domestic camels seem to diverge 0.4 or 0.3 mya, instead of the 0.5 mya stated in the main.

Response: We agree with this observation (Supplementary Fig. S15), and used 0.4 Mya (Line 282).

- For Fig S14, please add the coverage of the samples used for the G-PhoCS analysis. Only high coverage samples can be used for this analysis (and probably for PSMC). With only moderate depth (8-15x), many heterozygous sites remain undiscovered. This is essential for analyses relying on a single genome per population.

Response: We added the sequencing depth for the samples (Supplementary Fig. S16). They are all higher than average.

- In Fig S17, authors draw "domestication in Iran". I think they should say "domestication in Central Asia" instead, as they do in the main. They lack samples from other regions in Central Asia to claim that domestication occurred specifically in Iran.

Response: It is true that we can only conclude a Central Asian origin and we corrected the error (Supplementary Fig. S19).

- In the main, the migration bands inferred by G-PhoCS are reported as 1%. However, in table S13, it appears to be higher (2.225% and 2.170%). Please clarify.

Response: In Supplementary Table S13, m is just the migration rate per generation. The total migration rate per band $M = m\tau_m$, where τ_m is the time span of the band. We clarified this with footnotes of the table.

- *In the same table s13, the model with migration predicts that the split of dromedaries and wild animals occurred at the same time (35,300 ya)? Please justify this, as it could reflect a model mis-specification.*

Response: We re-run the MCMC and the split time could be distinguished.

- *Authors claim that "Although dromedaries were more divergent from both of the Bactrian camel species in phylogeny, the domestic Bactrian camels shared more variants with the dromedaries (52.69%) than with the wild Bactrian camel (31.14%) (Supplementary Figure S4) due to the tremendous reduction of genetic variants observed in the extant wild Bactrian camel.". Given their results, I think authors should add that: "... due to the tremendous reduction of genetic variants observed in the extant wild Bactrian camel and to gene flow between dromedaries and domestic Bactrian camels." (or something similar)*

Response: We added the suggested sentence (Line 148).

- *ADMIXTURE analysis reveals a bit of blue component (domestic Bactrian camels) in wild Bactrian camels. It appears to be significant at least for a few individuals. Based on TreeMix residuals, the donor population could be MG_TH. Authors should discuss about this possibility. F4 statistics are compatible with this. This is important given the endangered status of wild camels (perhaps not so wild then). The authors disregard this possibility and state that wild Bactrian camels have not exchanged genes with domestic livestock.*

Response: We discussed the possibility of introgression of domestic camels into wild

ones and highlighted the threat to the genetic distinctiveness of the wild species (Line 212-214, Line 231-232).

- *The authors state that "The differentiation among the domestic Bactrian camels was much lower, implying a single-origin of them.". Despite probably true, I feel the verb "implying" is too strong here. I do not think we can infer a single origin based on pairwise Fst values. The tree, where domestic Bactrian camels are monophyletic, is more conclusive. Please re-phrase to something as: "The differentiation among the domestic Bactrian camels was much lower, in line with their recent single origin".*

Response: We revised this sentence as suggested (Line 176).

REVIEWERS' COMMENTS:

Reviewer #1 (Remarks to the Author):

As this is my second time assessing this paper, please refer to my previous review regarding the main points and impression of the study - these have not changed.

Overall I am satisfied with the revisions made to the paper. Much clarity has been added to the methodology, specifically variant filtering and the table S13. The addition of credible intervals to the G-PhoCS estimates are also of great use to the reader, as they allow the high uncertainty of the estimates (particularly the very low lower-bounds of the divergence times) to be assessed and weighed together with the rest of the evidence present.

There is one major clarification I would ask to make to the text as it stands.

Line 363-364 is unclear - "those from KAZA continued to have a reduced increase in divergence from the dromedaries from other populations" - do the authors mean that "KAZA continued to have a reduced divergence from the dromedaries compare to other populations"? The authors refer here to Figure 3B, but from my reading of the Figure 3b heatmap their point it not supported. KAZA appears as diverged from Dromedary as other Bactrian groups. KAZA does appear to be the outgroup within "Non-Iranian dromedaries" in the tree, but the branch depth is extremely shallow. Could the authors clarify on this point - are they referring specifically to the tree? Do they mean "KAZA continued to have [reduced divergence] from the dromedaries from [compared to] the other populations"?

The authors should also clarify lines 365-366 - "This also supported an introgression of a ghost population into KAZA and it might not be excluded" . If the authors are referring to the tree in Figure 3b, then a bottleneck into east asia would also fit the observed pattern of genetic diversity - see Figure 1 for lower diversity estimates in East Asian populations compared to KAZA. The authors could clarify that the branching pattern in the Fig 3b tree does not contradict or is consistent with the "ghost admixture" model.

Minor recommendation - Line 311 - I recommend changing "animal domestication" to "livestock domestication" - dogs were likely domesticated prior to 11.5kya.

Overall the authors should be commended for the additional work made to the paper.

Reviewer #2 (Remarks to the Author):

After a few rounds of revisions, Ming and colleagues have satisfactorily addressed almost all my comments and suggestions, and I think the work deserves to be published as is.

The only comment that remains a bit ambiguous to me pertains to the split times inferred by G-PhoCS for the model with admixture. t_X and t_{root} are almost the same. I think the ghost was probably not sister to the root, but a different lineage. Not an issue though, because the authors openly discuss about the limitations of their model with introgression.

Responses to Reviewer #1

As this is my second time assessing this paper, please refer to my previous review regarding the main points and impression of the study - these have not changed.

Overall I am satisfied with the revisions made to the paper. Much clarity has been added to the methodology, specifically variant filtering and the table S13. The addition of credible intervals to the G-PhoCS estimates are also of great use to the reader, as they allow the high uncertainty of the estimates (particularly the very low lower-bounds of the divergence times) to be assessed and weighed together with the rest of the evidence present.

There is one major clarification I would ask to make to the text as it stands.

Response: We appreciate Reviewer #1's comments. Please find our responses to the concerns below.

Line 363-364 is unclear - "those from KAZA continued to have a reduced increase in divergence from the dromedaries from other populations" - do the authors mean that "KAZA continued to have a reduced divergence from the dromedaries compare to other populations"? The authors refer here to Figure 3B, but from my reading of the Figure 3b heatmap their point is not supported. KAZA appears as diverged from Dromedary as other Bactrian groups. KAZA does appear to be the outgroup within "Non-Iranian dromedaries" in the tree, but the branch depth is extremely shallow. Could the authors clarify on this point - are they referring specifically to the tree? Do they mean "KAZA continued to have [reduced divergence] from the dromedaries from [compared to] the other populations"?

Response: Yes, we are referring to the tree but not the heatmap of Figure 3B. We clarified the sentence according to the reviewer's suggestion (line 357-358).

The authors should also clarify lines 365-366 - "This also supported an introgression

of a ghost population into KAZA and it might not be excluded” . If the authors are referring to the tree in Figure 3b, then a bottleneck into east asia would also fit the observed pattern of genetic diversity - see Figure 1 for lower diversity estimates in East Asian populations compared to KAZA. The authors could clarify that the branching pattern in the Fig 3b tree does not contradict or is consistent with the “ghost admixture” model.

Response: As the last comment, we are referring to the tree of Figure 3B. We clarified that the branching pattern is consistent with the ghost admixture model for KAZA (line 358-359).

Minor recommendation - Line 311 - I recommend changing “animal domestication” to “livestock domestication” - dogs were likely domesticated prior to 11.5kya.

Response: We changed the word as recommended (line 303).